# Influences on PET Quantification and Interpretation

**DOI:** 10.3390/diagnostics12020451

**Published:** 2022-02-10

**Authors:** Julian M. M. Rogasch, Frank Hofheinz, Lutz van Heek, Conrad-Amadeus Voltin, Ronald Boellaard, Carsten Kobe

**Affiliations:** 1Department of Nuclear Medicine, Charité—Universitätsmedizin Berlin, Corporate Member of Freie Universität Berlin and Humboldt-Universität zu Berlin, 13353 Berlin, Germany; julian.rogasch@charite.de; 2Berlin Institute of Health at Charité, Universitätsmedizin Berlin, 10178 Berlin, Germany; 3Institute of Radiopharmaceutical Cancer Research, Helmholtz Center Dresden-Rossendorf, 01328 Dresden, Germany; hofheinz@hzdr.de; 4Department of Nuclear Medicine, Faculty of Medicine and University Hospital Cologne, University of Cologne, 50937 Cologne, Germany; lutz.van-heek@uk-koeln.de (L.v.H.); conrad-amadeus.voltin@uk-koeln.de (C.-A.V.); 5Department of Radiology and Nuclear Medicine, Cancer Center Amsterdam (CCA), Amsterdam University Medical Center, Free University Amsterdam, 1081 HV Amsterdam, The Netherlands; r.boellaard@amsterdamumc.nl

**Keywords:** positron emission tomography, quantitative accuracy, contrast recovery, signal-to-noise ratio, image interpretation, image quality

## Abstract

Various factors have been identified that influence quantitative accuracy and image interpretation in positron emission tomography (PET). Through the continuous introduction of new PET technology—both imaging hardware and reconstruction software—into clinical care, we now find ourselves in a transition period in which traditional and new technologies coexist. The effects on the clinical value of PET imaging and its interpretation in routine clinical practice require careful reevaluation. In this review, we provide a comprehensive summary of important factors influencing quantification and interpretation with a focus on recent developments in PET technology. Finally, we discuss the relationship between quantitative accuracy and subjective image interpretation.

## 1. Introduction

The purpose of this review is to provide a state-of-the-art overview of factors influencing common quantitative image parameters in positron emission tomography (PET) as well as image interpretation, which is usually not quantitative. To address this dichotomy, the chapter on “quantification” relates to factors with a bearing on quantitative accuracy, while the second chapter “interpretation” focusses on variables that affect the subjective, reader-dependent, mostly visual interpretation of images and their effects on diagnostic accuracy and response assessment. Some sections of the article put special emphasis on [^18^F]fluorodeoxyglucose (FDG), owing both to the unique clinical importance of [^18^F]FDG and its vast literature as well as the issue of dietary preparation and influences of blood glucose levels on quantification in [^18^F]FDG-PET.

PET quantification, as defined in this review article, comprises primarily those methodological factors that determine how accurately the radiopharmaceutical with its biodistribution in an individual patient is depicted. It focusses on those aspects that are potentially relevant for daily routine clinical care (Figure 1).

## 2. Factors Affecting PET Quantification

The essence of PET quantification is lesion contrast recovery (CR), which describes the relative recovery of the true focal activity concentration. Figure 2 illustrates the most relevant factors influencing lesion CR in PET.

### 2.1. Patient

The patient’s physiology and constitution influence the biodistribution of radiopharmaceuticals, especially of [^18^F]FDG, although they mainly affect standardized uptake values (SUV) in normal organs, such as the liver, brain, lung, skeletal muscles, and blood pool [1]. In a number of normal organs, the SUV corrected for body mass is positively correlated with the body mass index (BMI) [1,2,3]. This correlation can be partly explained mathematically by the incorrect estimation of the distribution volume in obese patients if total body mass is used instead of lean body mass (due to the relatively low [^18^F]FDG accumulation in fat tissue [4]) [5]. However, when compared to SUV in prostate-specific membrane antigen (PSMA)-PET, those in [^18^F]FDG-PET are more closely correlated with BMI [5]. This suggests effects on SUV in obese patients that are specific for [^18^F]FDG, possibly stemming from the positive correlation between BMI and blood glucose [3,6].

Mildly to moderately reduced kidney function (estimated glomerular filtration rate <60 ml/min) may not influence either normal organ SUV or blood pool clearance [6,7], unless kidney function is so far reduced that the patient requires hemodialysis [8]. Normal brain [^18^F]FDG uptake varies with age, sex, and the presence or absence of diabetes mellitus [1,6,9,10]. In women, [^18^F]FDG uptake in the ovaries and breast tissue varies with progesterone levels and age [11,12].

Notably, tumor SUVmax and SUVmean were not shown to vary systematically with the above-named physiological factors in a large meta-analysis of >20.000 individuals [1]. However, if tumor SUV is related to normal tissue SUV, these factors may have an indirect effect.

### 2.2. Patient Preparation

[^18^F]FDG uptake in tumors, inflammatory cells and brain tissue is mainly brought about by insulin-independent glucose transporters (GLUT) 1 or 3 [13,14,15,16,17,18,19]. Their metabolic rate (i.e., the absolute glucose uptake per time) is mostly constant at varying blood glucose levels, resulting in a lowering of absolute [^18^F]FDG uptake (i.e., SUV) by up to 50% if glucose levels are highly elevated. This is due to the competition of glucose and [^18^F]FDG for transporter molecules [6,20,21]. Conversely, raising blood glucose and/or insulin levels increases [^18^F]FDG uptake in the liver, skeletal muscles, and myocardium mainly via GLUT2 and 4 [1,6,22]. Physical exercise increases skeletal muscle [^18^F]FDG uptake via GLUT4 [23]. Therefore, to achieve optimal results for lesion-to-background ratios in [^18^F]FDG-PET, a low blood glucose level and insulin level should be ensured at the time of injection, and recommendations regarding physical activity should be observed [24].

Some drugs affect [^18^F]FDG uptake in normal organs. Examples of this are the variations in brain and cardiac uptake caused by sedatives [25,26] or elevated bone marrow uptake following administration of hematopoietic cytokines [27].

Regarding other tracers, specific recommendations to pause potentially interfering substances may also be important. The influence of somatostatin analogues on the uptake of neuroendocrine tumors in somatostatin receptor-specific PET [28] and of antihormonal treatment on PSMA uptake in prostate cancer cells [29,30] are currently under investigation. Further determinants of the biodistribution of various other radiopharmaceuticals are also beyond the scope of this article but have been addressed by previous reviews. These include physiological and pharmacological factors [31], the in vivo degradation of peptide-based PET radiopharmaceuticals [32], and interactions between the lesion microenvironment and radiopharmaceuticals [33,34,35,36].

Likewise, advances in kinetic modeling and parametric PET imaging [37] for the prediction and validation of pharmacokinetic models, e.g., in radioligand therapy [38] or in neurology [39,40], have been discussed elsewhere.

### 2.3. Partial Volume Effect

A general problem in the quantitative evaluation of PET data is the rather low spatial resolution. Even though improved scanner technology and reconstruction algorithms lead to a noticeably improved reconstructed spatial resolution compared to older systems (about 4 mm vs. 6 mm) [41], it is still much worse than the resolution of computed tomography (CT) or magnetic resonance imaging (MRI). The low resolution causes unavoidable partial volume effects, which can lead to a severe underestimation of tracer uptake, depending on the lesion size. Assuming a spherical lesion with homogeneous tracer accumulation and a spatial resolution approximated by a Gaussian point spread function (PSF) with 4 mm full width at half maximum (FWHM), the effect becomes relevant at a lesion diameter of about 10 mm. Here, the recovery of the maximum activity is still 97%, but it drops sharply with decreasing diameter, e.g., to 86% for an 8 mm diameter or 63% for a diameter of 6 mm (Figure 3) [42].

While these figures can easily be computed using the convolution of PSF and object geometry, they are of limited value in routine clinical practice since they apply only to the idealized situation described above. The recovery for lesions of a different shape and inhomogeneous tracer accumulation can differ widely from the figures presented [43], and it is therefore not possible to achieve reliable partial volume correction using these results.

Another factor besides spatial resolution that contributes to the partial volume effect is image sampling, meaning, in the context of PET, the distribution of a natural, irregular volume into cuboid voxels of a fixed size. Each voxel value then reflects the average activity in this voxel volume. In the case of lesion borders, some of the marginal activity is projected into the vicinity of the lesion (so-called spill-out; Figure 4A) while, conversely, activity at the lesion surface is diluted by background activity (spill-in). Furthermore, the activity within lesions is usually heterogenous due to numerous cell clusters of different tracer avidity. The projection of such clusters into comparably large voxel volumes with fixed size inevitably leads to attenuation of the true magnitude of heterogeneity and to a loss of spatial resolution of small heterogenous areas (Figure 4B). Consequently, PET systematically underestimates intralesional heterogeneity. The larger the voxel size, the higher the likelihood of underestimating both the true maximum activity within the lesion and the variety and heterogeneity of activities [42].

Both factors must be kept in mind when interpreting the SUV, especially SUVmax, of small lesions.

### 2.4. Hardware

#### 2.4.1. Silicon Photomultiplier (SiPM) Technology

The photomultiplier converts the photon emitted from the scintillation crystal into an electronic signal. Compared to conventional photomultiplier tubes (PMT), silicon photomultipliers (SiPM) in recently developed PET/CT scanners are smaller, provide up to 100% coverage of the scintillation crystal surface and offer high photon sensitivity, low noise, and fast timing resolution [44]. The chief advantages of SiPM are higher effective sensitivity and improved signal-to-noise ratios (SNR).

To estimate the independent added value of SiPM technology, performance metrics can be compared with otherwise similar PMT-based scanner models. Most importantly, these scanners should feature a similar length of the axial field of view (FOV) because the system sensitivity (cps/kBq), a main determinant of the SNR, increases quadratically with an increase in axial FOV (e.g., system sensitivity gain of approx. 80% for an axial FOV increase of 33%) [45]. The potential gain associated with SiPM is dependent on the relative coverage of the detector area with SiPM elements and their specific timing resolution relative to the PMT comparator. If, for example, a SiPM system with low relative coverage of <60% and a modest timing resolution of 382 ps [44,45] is compared to a bismuth germanate (BGO) PMT scanner with similar axial FOV length [46], the SiPM system’s sensitivity can be similar or even slightly inferior (20.8 vs. 23.3 cps/kBq) [45]. In contrast, a SiPM system with 100% detector coverage and 214 ps timing resolution [47] can offer increased system sensitivity over a comparable PMT system with lutetium oxyorthosilicate (LSO) crystals [48] that exceeds the expected difference based on the slightly larger axial FOV (16.4 vs. 9.6 cps/kBq with a 25.6 vs. 22.1 cm FOV) [47]. If further improvements in timing resolution are achieved with future generations of SiPM [44], this could bring with it a proportional increase in effective sensitivity (noise equivalent counts; NEC) [49,50]. Such a gain in counts could translate into an SNR gain that is equivalent to the square root of the relative improvement in timing resolution [49,51].

Additionally, the development of small SiPM elements has enabled small voxel sizes down to 1 × 1 × 1 mm^3^ [52] with potential superiority over PMT-based systems in reconstructed spatial resolution and quantitative accuracy in the case of small lesions [53,54,55,56].

#### 2.4.2. PET/CT vs. PET/MRI

CT-based attenuation correction (CT-AC) for PET has the advantage of directly and reliably translating tissue densities from CT Hounsfield units (e.g, acquired with 120 keV) into linear attenuation correction factors for 511 keV photons [57]. In contrast, MR-based attenuation correction (MR-AC) in PET/MRI relies on different, vendor-dependent MR sequences that are used to automatically segment the body into three or four compartments (air, bone, soft tissue, and fat) [58]. These compartments are assigned specific linear attenuation correction factors for 511 keV photons. However, several sources of error exist, including extracorporeal components such as headphones, dense hair or MR coils [59], incorrect attenuation correction factors for bone and lung tissue [60], incorrect segmentation of such compartments, and MR image truncation [61]. Systematic underestimation of SUV in bone tissue or lesions close to the bone of >10% compared to PET/CT has been described [60,62], whereas SUV differences were <5% in other normal tissues [60]. It should also be noted that cross-calibration of PET/MRI scanners is generally hampered by water-filled phantoms, which are commonly used for SUV calibration in PET/CT and can produce substantial artifacts in the MR-AC sequence with corresponding SUV errors [63,64]. In such phantoms, inter-vendor differences in MR-AC maps [61] can lead to SUV differences in phantom spheres of 10–20% [58].

Studies that have investigated normal organ and lesion SUV in patients undergoing both PET/CT and PET/MRI after a single injection of [^18^F]FDG were mostly biased by a non-randomized scan order. Lesion SUV were usually lower on the first examination, which can partly be explained by the increasing [^18^F]FDG accumulation in most (malignant) lesions beyond 60 min after tracer administration. However, systematically different normal organ SUVs have been reported for [^18^F]FDG as well, which most likely result from MR-AC inaccuracies [60,65,66]. Furthermore, as such studies can only compare one single PET/CT scanner with one PET/MRI scanner and have usually only investigated one specific set of reconstruction parameters, it is difficult to draw a generalized conclusion regarding the comparability of SUV in clinical PET/CT and PET/MRI.

### 2.5. Quality Control

High image quality is of general importance, but special requirements are necessary in the case of SUV quantification. Three quantities are needed for the computation of SUV: the injected activity, the activity concentration in the lesion as determined in the image data, and the distribution volume of the radiopharmaceutical (usually approximated by the body mass of the patient). Errors in these quantities lead directly to errors in the SUV of the same magnitude; e.g., a 10% error in the patient body mass leads to a 10% error in SUV. An accurate determination of the patient’s body mass is therefore indispensable for quantitative purposes and of special importance if follow-up studies are to be compared.

The accuracy of the measured target activity concentration depends on accurate cross-calibration of the scanner with the dose calibrator that is used to measure injected activities. Frequent verification is therefore mandatory, and the European Association of Nuclear Medicine (EANM) recommends quarterly cross-calibrations [24]. To determine the injected net activity, the activity in the syringe has to be measured before and after injection. Furthermore, since the activity concentration in the lesion and the injected activity are measured with different devices, correct decay correction requires the two devices to have synchronized clocks. The optimal way to achieve this is by connecting both devices to the same time server. If this is not possible, regular verification of a synchronized time is necessary. Although all these errors are potentially severe, it should be noted that they can easily be avoided by following existing guidelines published by the EANM [24] or the EANM Research Ltd. (EARL) initiative [67].

### 2.6. Acquisition Parameters

#### 2.6.1. Uptake Time after Tracer Administration

Lesional uptake of [^18^F]FDG is more or less irreversible in neoplastic cells [68,69] due to intracellular trapping of the phosphorylated molecule, although reversible uptake has been demonstrated in inflammatory tissue [18]. The SUV of tumor lesions, therefore, increases steadily over time after [^18^F]FDG injection [70,71]. Normal organ SUV may either decline over time (blood pool, bowel, and fat tissue), remain relatively stable (liver and lung) or increase (cerebellum, spleen, bone marrow, and muscles) [72,73]. Therefore, to ensure the repeatability of SUV measurements, a similar uptake time should be observed [24]. It should be noted that tumor uptakes of [^68^Ga]Ga-PSMA-11 [74] and the somatostatin receptor-specific agents [^68^Ga]Ga-DOTATOC and -DOTATATE [75] have recently also been described as irreversible.

#### 2.6.2. Acquisition Duration per Bed Position

SUV are corrected for injected activity and acquisition time per bed position. Due to this correction, they are less directly affected by changes in these variables than image noise (Section 3.2.1). This is especially true for SUVmean and, to a lesser degree, for SUVpeak, which are both relatively stable even after reduction of acquisition time to 50% or less [76,77,78,79,80]. In contrast, SUVmax can vary substantially because this is, by definition, the outlier, which is most affected by increasing relative errors with declining count statistics [76,80]. However, systematic SUVmax increases of >5–10% have not typically been observed at acquisition times per bed position of >30–60s, at least during investigations of SiPM-equipped systems and/or patients with BMI <25 kg/m^2^ [79,81,82].

#### 2.6.3. Respiratory Motion Correction

As static PET images require considerable acquisition time for each bed position, a bed position covers several breathing cycles, and the final image represents the average of detected activity in each location. The true signal in organs and lesions that are subject to respiratory motion is thereby distorted and blurred along the vector of motion. This results in lower maximum activity but higher apparent lesion volume of the target lesion at an equal relative SUV threshold. This has been reported, e.g., for pulmonary lesions (especially in the lower and middle parts of the lung), liver lesions, and pancreatic lesions [83,84,85,86,87,88]. Several techniques for respiratory motion correction have been proposed [89].

In cardiac PET imaging, contraction of the myocardium further contributes to the quantitative inaccuracy of uncorrected static PET protocols [90,91].

### 2.7. Image Reconstruction

The effect of image reconstruction algorithms on quantitative accuracy in PET has been studied extensively, and three recent reviews cover current knowledge and views on time of flight (TOF) integration [44] as well as PSF modelling and Bayesian penalized likelihood (PL) reconstruction [92,93].

Briefly, TOF increases the lesion’s CR and SUV compared to non-TOF PET at a comparable level of image noise [94,95,96]. This effect is especially prominent in low-contrast lesions [95].

PSF reconstruction, or resolution modelling, refers to compensation for the scanner’s specific PSF throughout the transaxial FOV as part of the reconstruction process. This improves reconstructed spatial resolution [97,98] and increases lesion SUV [99,100] but can lead to overestimation of the true activity due to so-called Gibbs’ artifacts [92,93]. Compared with non-PSF reconstructed images, this can increase lesion SUVmax, SUVmean and SUVpeak by up to 30% on average [100,101,102]. To correct for these increases, an additional Gaussian filter can be applied during image reconstruction or to the final images [100,102]. With appropriate filter width (FWHM) based on sphere CR from standardized phantom measurements, PSF-induced SUV increases can be negated, resulting in comparable lesion SUVs to those in non-PSF data. Kaalep et al. showed that by filtering PSF-reconstructed data that were compliant with the updated EARL2 standard, SUV and metabolic tumor volumes (MTV) in lung cancer and lymphoma lesions could be achieved that were similar to non-PSF data (EARL1-compliant) [102]. Houdu et al. demonstrated that prognostically relevant SUVmax thresholds in patients with lung cancer are only valid in data reconstructed in compliance with the same standard as the dataset that defined this prognostic threshold [103]. A harmonization of PET data is therefore recommended when quantitative data are to be analyzed from different PET systems.

Bayesian PL reconstruction is an iterative method that employs the Bayesian principle of integrating estimates about the physical properties of the unknown image as a prior probability with the aim of improving its prediction [104]. Furthermore, a penalization/regularization term *beta* is included that penalizes large intensity differences between neighboring voxels and thereby aims at controlling the noise and Gibbs’ artifacts. The beta factor, which is user-defined, determines the weight (importance) of this penalty [105,106]. Using a commercially available PL reconstruction (General Electric [GE] Q.Clear), several phantom studies have shown that PL reconstruction can increase the CR of standardized sphere inserts compared to conventional TOF and PSF reconstruction [41,46,101,107,108]. Although this effect can lead to overestimation of the true activity in larger lesions if the SUVmax is used [101], the increase in CR may be especially prominent in microspheres with diameters <10 mm, in which conventional algorithms usually underestimate the true activity substantially [109]. This has been confirmed by increasing lesion SUVmax compared with TOF and PSF reconstruction, particularly in small pulmonary lesions [110,111]. However, this difference is directly dependent on the user-defined penalization factor beta during PL reconstruction, and at beta values of 300 to 600, which have been rated as optimal for visual reading of [^18^F]FDG-PET images by human readers [107,112,113], inter-method SUV differences may no longer be significant [41,108,111].

Besides these clinically established algorithms, several reconstruction algorithms based on artificial intelligence, namely deep learning techniques, have recently been proposed [114].

Any current PET image reconstruction algorithm includes correction for scatter, randoms, dead time, and attenuation. Regarding CT-AC, the presence of an intravenous contrast agent in the target tissue results in overestimation of attenuation and, therefore, higher SUV. In tumor lesions, such increases are usually <10% [115,116,117] and have been deemed irrelevant for visual assessment in previous studies [118,119]. However, in organs with a particularly high concentration of the contrast agent (e.g., the liver, kidney, or blood vessels), these deviations can be higher [116,120,121]. Thus, using a non-contrast-enhanced CT for attenuation correction is recommended when quantification by SUV is planned [24].

## 3. Factors Affecting PET Interpretation

Interpretation of PET images aims at classifying lesions or tissues according to their differential diagnosis at a single time point or at evaluating changes in lesion or tissue biology over time. Both may contain prognostic or predictive information.

Figure 5 presents the most relevant factors influencing PET interpretation. Selected factors are discussed in the following respective sections.

### 3.1. Specificity of the Radiopharmaceutical

In any thorough examination of the factors confounding quantitative accuracy, it should be kept in mind that the appropriateness of the radiopharmaceutical to assess the tissue or lesion characteristics of interest may be of utmost importance to the reader’s certainty and correctness in interpreting PET images. If the radiopharmaceutical does not allow the classification of a lesion on a biochemical basis, e.g., the differentiation of a malignant or benign cause, the achievement of quantitative accuracy will not be helpful or relevant.

This becomes most evident with [^18^F]FDG, which is specific neither to malignant lesions nor to discrete tumor entities. In oncology, this hampers the differentiation between inflammatory changes and neoplastic tissue [15,18] or benign lesions and well-differentiated malignant lesions with low [^18^F]FDG avidity [122,123,124,125]. Various radiopharmaceuticals with higher tumor specificity have therefore been developed to increase diagnostic accuracy for certain tumor entities, e.g., [^68^Ga]Ga-PSMA-11 or [^18^F]F-PSMA-1007 in prostate cancer [126,127], somatostatin receptor-specific tracers for neuroendocrine tumors [128], radiolabeled peptides in brain tumors [129] or [^18^F]fibroblast activation protein inhibitor (FAPI) for different carcinoma types [130]. However, sources of false positive or negative findings still remain with these tracers [131,132,133,134] and must be considered during image interpretation.

In cardiovascular imaging with [^18^F]FDG-PET, insufficiently suppressed physiologic [^18^F]FDG uptake by the myocardium can complicate the differentiation from inflammatory changes [135], while postoperative changes or sterile inflammation can be difficult to differentiate from active infection [136,137,138]. Alternative tracers that are more specific for inflammation [139,140] or bacterial infection [141] might facilitate interpretation.

### 3.2. Image Quality and Lesion Detection

In the visual assessment of PET images in routine clinical practice, quantitative accuracy cannot usually be directly assessed because the ground truth is unknown. However, the subjective, perceived image quality can be rated, and quantitative measures can be used to derive an objectified surrogate for image quality. In this sense, a maximized contrast-to-noise ratio (CNR) reflects high image quality [142], because high lesion CR and low background noise are both key to achieving high diagnostic accuracy (i.e., to minimize false-negative and false-positive results). Therefore, all previously discussed factors on CR and image noise have a direct influence on image quality.

#### 3.2.1. Injected Activity and Acquisition Time

Subjective image quality is affected by the relationship between injected activity per kilogram body mass and acquisition time per bed position. A low product of the two factors results in excessive image noise and possibly increased rates of false-positive results and decreased reader confidence [76,143,144]. The EANM, therefore, recommends a minimum of 7 MBq/kg*min for [^18^F]FDG-PET using a contemporary PET system with >30% overlap between bed positions [24]. Alternatively, a formula that includes the quadratic weight can be used, which could better compensate for loss in image quality in patients >75 kg [24]. Moreover, EARL also provides a procedure that can be followed to determine a lower activity prescription for systems with very high sensitivity or improved timing resolution (e.g., <300 ps) [145].

The anticipated benefits of PET hardware and software improvements over the last decade are perhaps reflected in figures for the lower minimum of injected activities required for state-of-the-art PET systems. Using an older non-TOF PET scanner with BGO crystals and 15.7 cm axial FOV [146], Geismar et al. recommended 10 MBq/kg*min for [^18^F]FDG-PET in patients with a BMI >22 kg/m^2^, while 8 MBq/kg*min was recommended only in patients with BMI <22 kg/m^2^ [144]. Using a modern SiPM-equipped PET scanner with 20 cm axial FOV and PL reconstruction [54], Trägårdh et al. proposed 6 MBq/kg*min for [^18^F]FDG-PET to achieve acceptable image quality and lesion visibility [76]. Moreover, the authors recommended 8 MBq/kg*min for [^18^F]F-PSMA-1007 based on the same scanner and PL reconstruction [143].

Wickham et al. [147] investigated the relationship between subjective and quantified measures of image quality in 111 clinical [^18^F]FDG-PET/CT scans (oncology, hematology, and infection) using a PMT-based PET scanner with a 22.1 cm axial FOV [148]. The optimal formula to predict high image quality included sex (higher activity in women), body mass and height. Neither patient age nor normalized body metrics or different, more sophisticated measures of body tissue composition provided added value in predicting image quality [147].

However, although a standardized measure of image quality is used, the studies cited here are still not directly comparable, as the axial FOV length differed substantially. To address this problem, the FOV length or–more accurately–the system’s sensitivity in cps/kBq would have to be included in the formula to calculate the required injected activity. This becomes especially evident with recent PET systems with extra-large axial FOV of >25 cm and their potential to substantially reduce required acquisition times [47,149,150].

Furthermore, some publications have relativized the general assumption that injected activity and acquisition time are linearly interchangeable when aiming at constant image noise. These studies have demonstrated that image quality (namely image noise) in overweight patients >80–90 kg benefits especially from increased acquisition time per bed position [77,151,152].

#### 3.2.2. SiPM Technology

The increase in effective sensitivity enabled by modern SiPM could be translated into an equivalent reduction in injected activity or acquisition time while retaining equal image quality as PMT [49,51]. Intraindividual comparisons of clinical PET data between scanners equipped with either SiPM or PMT are scarce because this requires a second scan in a randomized protocol to prevent bias from systematically different uptake times between the scans. Sekine et al. used a randomized protocol to investigate the potential to reduce injected [^18^F]FDG activity required (or acquisition time) through the use of an SiPM TOF PET/MRI instead of a PMT TOF PET/CT in 74 patients with different types of malignant tumors. Image quality (artifacts, noise, and sharpness) was rated as acceptable at up to 40% reduction in simulated acquisition time with the SiPM PET/MRI. However, the potential to reduce acquisition time with the PMT PET/CT was not specifically investigated. Moreover, the SiPM PET/MRI had a 25 cm axial FOV compared to the 15.7 cm FOV of the PET/CT system, which partly explains differences in image quality (especially image noise) independent of the photomultiplier technology [153].

López-Mora et al. reported higher lesion detection rates with a SiPM PET/CT compared to a PMT PET/CT in 22 of 100 patients using [^18^F]FDG or [^18^F]fluorocholine (58 patients underwent SiPM PET first; axial FOV were similar). This resulted in a modified disease stage in 7 of these 22 patients based on SiPM PET/CT [154]. Similarly, Baratto et al. found higher lesion detection rates in 13 of 94 patients with [^68^Ga]Ga-DOTATATE using SiPM vs. PMT PET/CT in randomized order (SiPM, 20 cm axial FOV; PMT, 15.7 cm) [155].

Consequently, SiPM technology may well produce improvements in image quality and lesion detection rates. However, the magnitude of improvement that can be achieved in clinical scans is likely to vary with different SiPM designs.

#### 3.2.3. Image Reconstruction

Among the technical factors that affect PET interpretation, image reconstruction is especially influential because it directly affects both image noise and lesion CR to a potentially high degree (Figure 6).

Compared to standard ordered subset expectation maximization (OSEM) reconstruction, OSEM with TOF shows improved noise characteristics [49,94,156,157,158]. Surti et al. showed that TOF reconstruction improves detection rates of simulated liver and lung lesions by human readers. This improvement was pronounced in heavy patients with BMI ≥ 26 kg/m^2^ [159]. The same group further demonstrated that TOF improved lesion detection, especially in low-contrast lesions [160].

PSF compensation can also increase the SNR [157,158] and thereby the subjectively rated image quality [157,161,162]. Investigations with an anthropomorphic phantom or with simulated liver and lung lesions have shown that TOF and PSF can have a supplementary effect on increasing lesion detection rates [161,163]. Conflicting results were reported by other authors who did not find higher lesion conspicuity or detection rates with PSF in small patient samples [162,164,165]. These discrepancies may stem from the fact that the ability of PSF to increase lesion CR is most prominent at the periphery of the transaxial FOV [98,166,167] and in small, high-contrast lesions [95], such as pulmonary nodules. This is illustrated by the observation of Schaefferkoetter et al. of an improvement in lesion detection rates with PSF limited mainly to the lung, while the detection rates achieved with TOF extended to the liver and lung [161].

Based on the potential of PL reconstruction to systematically improve SNR compared to OSEM-based algorithms [108,113,168], several authors have reported improved SNR and image quality with different radiopharmaceuticals, such as [^18^F]FDG [107,112,113], [^18^F]F-PSMA-1007 [143], [^68^Ga]Ga-PSMA-11 [169,170], [^68^Ga]Ga-DOTATATE [171] or ^89^Zr-labelled tracers [172]. PL reconstruction has repeatedly been shown to increase conspicuity and detection rates of pulmonary lesions, even compared to OSEM with PSF and/or TOF [112,113,164,173,174]. Figure 7 shows a case example.

In contrast, in a small sample of 13 patients undergoing [^18^F]fluorocholine PET/CT for prostate cancer staging, PL with different beta values showed a comparable number of positive lymph nodes to that revealed by OSEM with PSF and TOF [175].

When estimating diagnostic accuracy from reported lesion detection rates, it is important to recognize that there is usually no gold standard available with which to assess the correctness of detected lesions and that such analyses are therefore unable to evaluate specificity. As an exception, Teoh et al. retrospectively investigated the diagnostic accuracy of OSEM + TOF and PL reconstruction using SUVmax and visual reading in 121 pulmonary nodules. Here, histological verification was available. Diagnostic sensitivity and accuracy were similar with both algorithms, while specificity tended to be lower with PL than with OSEM + TOF, especially in lesions >10 mm diameter [173].

Furthermore, no blanket conclusion on differences in image noise or lesion detection between reconstruction algorithms should be drawn from isolated results comparing two algorithms with only one set of parameters each (e.g., number of iterations or type of in-plane filter). Such parameters, namely the in-plane filter width or, in the case of PL reconstruction, the beta value, can have drastic effects on image noise and lesion CR (Figure 2). A higher filter width or beta value systematically decreases both image noise and CR. Reconstruction algorithms should therefore be compared with multiple sets of parameters to investigate real systematic differences between the methods [108]. It may otherwise be observed that such differences can only be detected under specific conditions [41,175].

In a study on 52 patients with lymphoma, 5 patients undergoing [^18^F]FDG-PET for restaging were divergently classified as non-responders (Deauville score 4–5) with PL reconstruction but as responders (Deauville 1–3) with OSEM (without TOF or PSF; compliant with the EARL1 standard) [176].

### 3.3. Relationship between Objective and Subjective Image Quality

Although CNR, SNR and NEC are surrogates for image quality, none of these single parameters sufficiently reflects subjective image quality as a whole [147,177]. However, adequately defined quantitative assessments may each measure specific aspects of subjective image quality, such as image sharpness, lesion contrast or image noise [178].

Several studies on subjective image quality in whole-body [^18^F]FDG-PET with PL reconstruction found that image quality was highest at beta values of 450 (to 600) despite lower lesion CR or “image sharpness” at these beta values compared to lower values [41,108,112,175,179]. This confirms that subjective image quality is a combination of lesion contrasts and image noise and that readers may demand adequately low noise levels even if this comes at the expense of lesion CR (i.e., quantitative accuracy). In low-count conditions, this tendency to prefer smooth, low-noise images with beta values >600 over “sharper” images could become even more evident [112,170]. Thus, images that are rated best regarding subjective image quality are not necessarily those with the highest quantitative accuracy. Conversely, Zhang et al. reported that lesion SUVmax and detection rates in [^18^F]FDG-PET remained significantly unchanged despite decreasing acquisition time per bed position from 900 s to 60 s and steadily decreasing subjective image quality [180]. Quantitative accuracy may therefore not necessarily require optimal (subjective) image quality. As these criteria may not be equally fulfilled by a single reconstructed dataset, a reconstruction of separate datasets has been proposed for visual reading or optimized quantification in routine clinical practice [24,181,182].

### 3.4. Relationship between PET Quantification and Image Interpretation

#### 3.4.1. Quantitative or Visual Interpretation Criteria?

Interpretation of PET images in routine clinical practice primarily follows visual criteria, i.e., the assessment of generalized or focal pathologies in tracer accumulation, while quantitative parameters, including SUV, provide orientation or additional information at most [24]. As the use of SUV to quantify tracer uptake increased, it was anticipated that this would represent a standardized, reliable criterion to classify lesions with their biological properties and prognostic implications. Consequently, diagnostic SUV thresholds have been proposed for pulmonary nodules [183], lymph node staging in lung cancer [184], adrenal lesions [185], musculoskeletal tumors [186], tumor delineation in gliomas [187], and response assessment in lymphoma [188] among other things. Thus, it is reasonable to assume that the achievement of quantitative accuracy will bring certainty and correctness to lesion interpretation.

However, lesion SUVs in [^18^F]FDG-PET show a test-retest variability in the same patient with the same PET scanner of up to 20% [189] and are usually even less comparable between different scanners and centers or under routine clinical conditions [190]. This has undermined any attempts to establish widely adoptable SUV thresholds unless rigorous harmonization measures are followed [101,103,181]. Given the inability to derive generalizable SUV thresholds, it has not yet been possible to prove that SUV or any other quantitative measures used in clinical practice provide additional value over visual assessment alone for routine clinical diagnostics [24].

#### 3.4.2. SUV: Which Parameter?

If SUV measurements are taken to support the visual assessment of PET images in routine clinical practice, this probably occurs most often in assessments of the response to therapy. However, as stated above, the validity of these measures is determined by the test-retest variability. Despite the common use of SUVmax in clinical practice, arising from its convenience, SUVpeak and SUVmean have been shown to be slightly less variable under test-retest conditions [189,191]. However, SUVmean and SUVpeak are affected by the reproducibility of the size and placement of the volume of interest (VOI) [192], which requires appropriate standardization or automation. Consequently, the choice of SUV parameter can fundamentally change the assessment of disease progression or response to treatment in the majority of cases [193]. A consensus was therefore needed, and Wahl et al. proposed the PET Response Criteria In Solid Tumors (PERCIST 1.0) in 2009 with the aim of standardizing the SUV parameters (SULpeak = SUVpeak corrected for lean body mass), the VOI size, the definition and number of appropriate target lesions and thresholds for response categories [194]. Still, the repeatability of the liver SULmean under clinical conditions in the same patient during treatment has been shown to be only fair (intraclass correlation coefficient <0.6) [195]. Consequently, the use of SUV to support valid clinical response assessment outside of study conditions remains highly challenging.

#### 3.4.3. MTV: Which Delineation Method?

Treatment decisions based on clinical risk stratification might be further improved by including the initial tumor volume in [^18^F]FDG-PET, as this factor has been shown to be an independent prognostic value for patient survival in conditions such as non-small cell lung cancer [196,197], different gynecological malignancies [198,199,200,201], and head and neck cancer [202,203]. Initial results also show a prognostic value of PSMA-PET tumor volume in prostate cancer prior to radioligand therapy with [^177^Lu]Lu-PSMA-617 [204].

However, measurement of tumor volume is not yet a standardized procedure because numerous methods have been described to delineate the tumor volume, and considerable differences have been reported between those methods [205,206]. The most convenient and common approaches range from the use of fixed absolute or relative activity thresholds to adaptive methods based on the local signal-to-background ratio. Consequently, optimal volume thresholds to separate prognostic groups may differ systematically and foster discordant assessments, although with optimized thresholds, each method on its own may retain its prognostic value [207,208,209]. Therefore, for both scientific and clinical use, tumor volume should be calculated by parameters that are readily available and promise high reproducibility between different readers and institutions.

### 3.5. Inter-Reader Variability

Inter-reader or -rater variability can limit the reliability of PET imaging (Figure 5). Numerous studies have analyzed the inter-reader agreement of visual PET assessments, particularly in [^18^F]FDG-PET for lymphoma and PSMA-PET for prostate cancer. In lymphoma patients, several studies have reported high inter-rater agreement using the standardized Deauville criteria [210,211], although conflicting results have also been reported [188,212]. It has been shown that reader training and discussions over divergent assessments increase agreement even among “expert” readers [213]. It has been suggested that SUV-based criteria in lymphoma might improve inter-reader agreement because they are unaffected by visual contrast effects [188]. However, both SUV measurements and visual Deauville criteria may be affected by image reconstruction, such as PL reconstruction [176]. Furthermore, besides the reader’s subjective assessment of a certain lesion, in a setting in which several lesions of interest are present (e.g., in restaging in lymphoma or metastatic tumors), additional inter-reader variability can result from a divergent choice of the decisive target lesion [213].

A systematic comparison of inter-reader agreement based on quantitative measures and based on visual reading has rarely been performed, and there is still little evidence of any additional value of the quantitative approach. Furthermore, any quantitative criteria and diagnostic thresholds are a result of certain methodological and technical conditions, which may change over time and require adaptation. However, it has been demonstrated that inter-rater agreement in response assessment with [^18^F]FDG-PET in non-small cell lung cancer or metastatic breast cancer can be considerably improved through the use of the target lesion SULpeak (PERCIST 1.0 criteria) in comparison with a subjective assessment [194,214,215].

Regarding PSMA-PET for prostate cancer, several standardized evaluation criteria have recently been proposed [216,217,218,219] with the aim of improving inter-reader agreement [219] and to aid inexperienced readers [220]. However, inter-reader concordance remains higher between experienced readers [221] and, depending on the specific diagnostic task, substantial to almost perfect agreement has usually been reported [220,221,222,223]. Similar degrees of inter-reader agreement were achieved with different reporting criteria [224]. However, standardized reading criteria do not negate the dissimilarities in the images obtained using different types of PET hardware and methods of image reconstruction, and both SiPM technology [79] and PSF reconstruction [225] have been found to result in systematically higher lesion conspicuity despite standardized reporting criteria (PSMA-RADS) [216].

## 4. Conclusions and Perspectives

As we have demonstrated here, a variety of factors influence PET quantification and interpretation. All variables should be considered potential sources of error when interpreting clinical PET images. Although the added value of quantitative uptake parameters for clinical decisions is still not well-defined, it should be kept in mind that even simple quantitative measures such as the SUV are highly variable. The emergence of new PET technologies such as SiPM and advanced image reconstruction algorithms further contributes to the complex issue of image quality and quantitative accuracy. Stringent measures of quality control and standardized imaging protocols should therefore be implemented to ensure robust and valid imaging results in routine clinical care. This will also be crucial to explore and validate the clinical utility of machine learning-based image biomarkers. To ensure comparability, we recommend adhering to the procedure guidelines by the EANM. Furthermore, the EARL initiative has proposed standards for a systematic standardization between imaging centers. This may include the reconstruction of different data sets for image interpretation: (1) a data set for optimal visual lesion detection and (2) a data set for standardized and quantitative image interpretation.

## Figures and Tables

**Figure 1 diagnostics-12-00451-f001:**
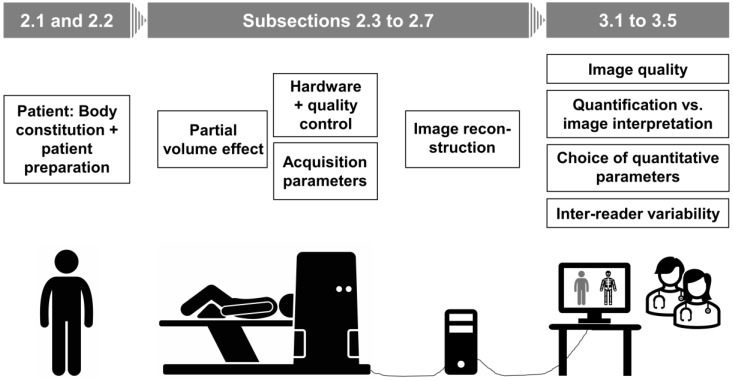
Article structure: From patient preparation to image interpretation. Every step of preparing the patient, acquiring and processing PET images, and choosing criteria to quantify and interpret the data potentially affects quantitative and diagnostic accuracy. Each of these steps is addressed by successive subsections of this article.

**Figure 2 diagnostics-12-00451-f002:**
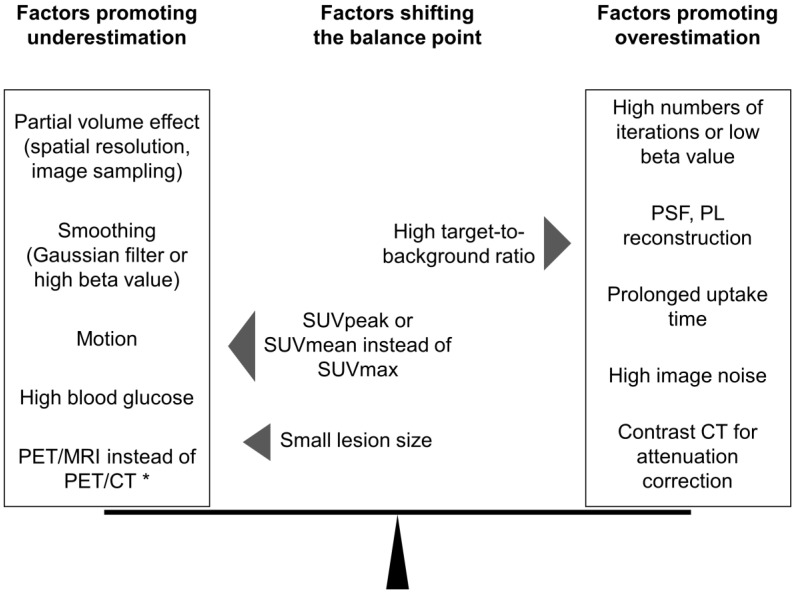
The thin line of quantitative accuracy in PET. Quantitative accuracy of PET in lesions (i.e., recovery of the true activity concentration) can be imagined as a pair of balances between factors that promote either under- or overestimation of the true activity. Additionally, the point at which the combination of these contrasting factors achieves quantitative accuracy is influenced by lesion-specific and methodological factors (e.g., the choice of standardized uptake value (SUV) parameter). ***** Reported lesion SUVs in PET/MRI are lower than those in PET/CT; however, this may not be true for every lesion in every tissue. PSF, point spread function; PL, penalized likelihood.

**Figure 3 diagnostics-12-00451-f003:**
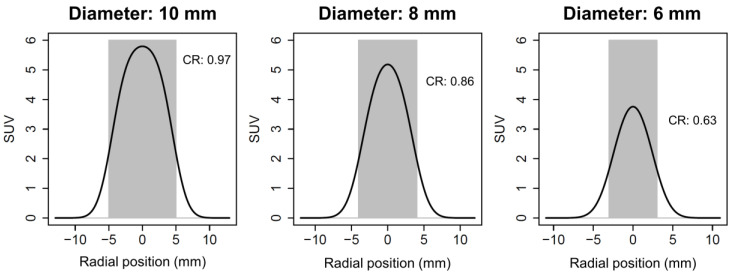
Reconstructed spatial resolution. The effect of limited reconstructed spatial resolution on lesion contrast recovery (CR) at different lesion diameters is demonstrated here, while the additional effect of image spacing is disregarded. In this example, spatial resolution is always 4 mm full width at half maximum (FWHM). The true lesion activity is shown in light grey, and the displayed activity is shown by the black line. In a 10-mm lesion (left), CR is 0.97, which is close to the optimum of 1.0 but decreases considerably with decreasing lesion diameter despite equal true activity. Please note that these values are calculated for lesions with absent background activity. If background activity is present, relative CR increases systematically.

**Figure 4 diagnostics-12-00451-f004:**
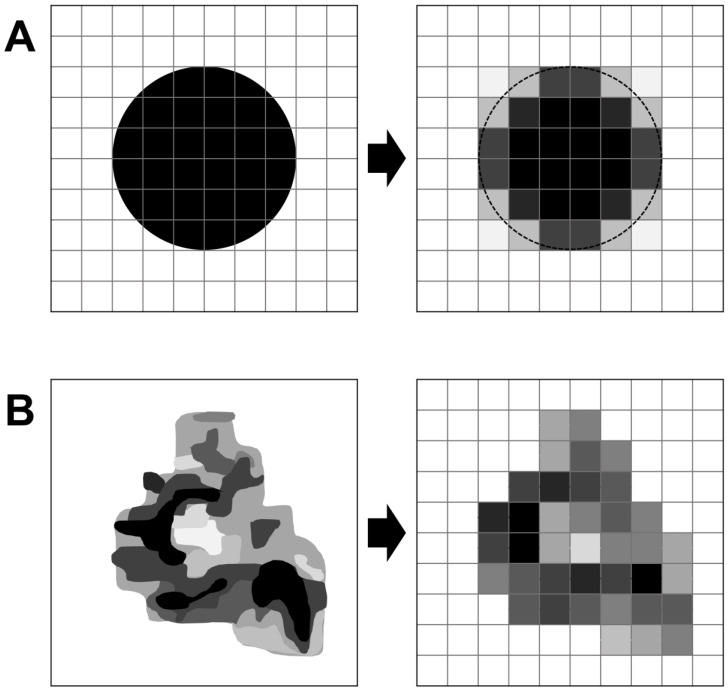
Image spacing. The effect of image spacing is illustrated in two dimensions (but would have to be extrapolated to three dimensions for PET data). (**A**) An idealized homogenous, spherical lesion is displayed on the left side with the superimposed voxel grid. The depiction of this lesion is shown on the right. At the lesion border (surface), image spacing results in a dilution of lesion activity by background activity (spill-in). In the usual case of a hot lesion, this spill-in leads to underestimation of the average lesion activity. Conversely, some of the marginal lesion activity may be visualized outside of the true lesion border (spill-out), and the lesion may appear larger than it truly is (dotted line). (**B**) In a lesion with heterogeneous activity (illustrated by different grey values), image spacing leads to an underestimation of intralesional heterogeneity because each voxel only represents an average activity, and both maximum and minimum intensities are attenuated. The minimum spatial resolution is determined by the voxel size. In (**B**), the effects of spill-in and spill-out at the lesion border are disregarded for simplification.

**Figure 5 diagnostics-12-00451-f005:**
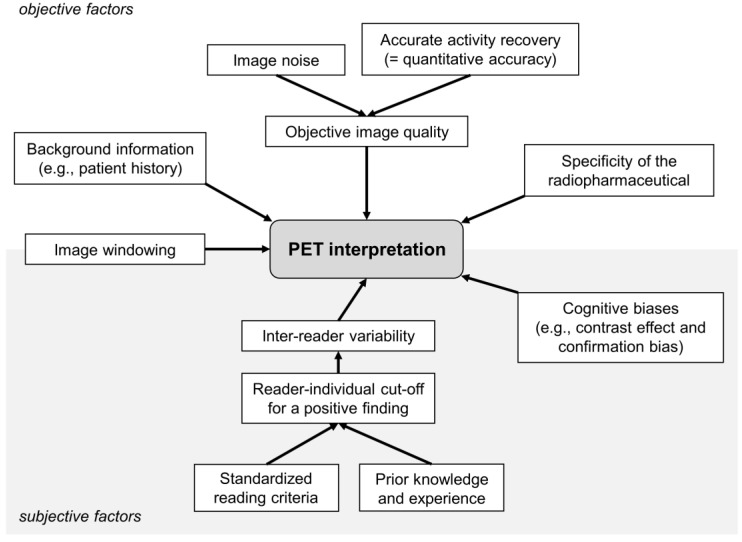
Factors affecting PET interpretation.

**Figure 6 diagnostics-12-00451-f006:**
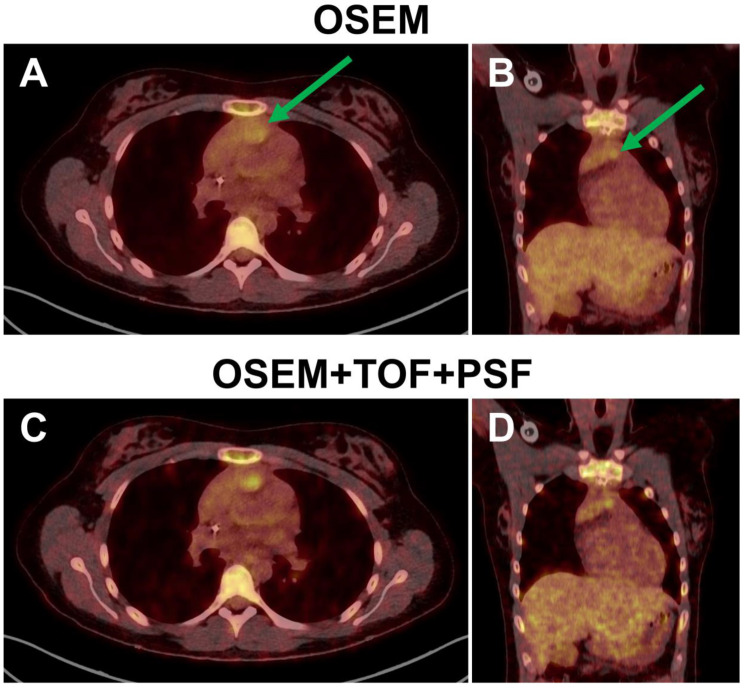
Fused transaxial as well as coronal PET/CT slices through residual mediastinal lymphoma tissue of a 23-year-old female patient reconstructed with the OSEM algorithm (**A**,**B**) and with OSEM combined with TOF and PSF (**C**,**D**). While the lesional [^18^F]FDG uptake was defined as Deauville score 3 based on OSEM reconstruction, it would exceed the liver uptake when assessed based on images reconstructed with TOF and PSF (=Deauville score 4). This could alter the assessment from “adequate” to “inadequate” metabolic response.

**Figure 7 diagnostics-12-00451-f007:**
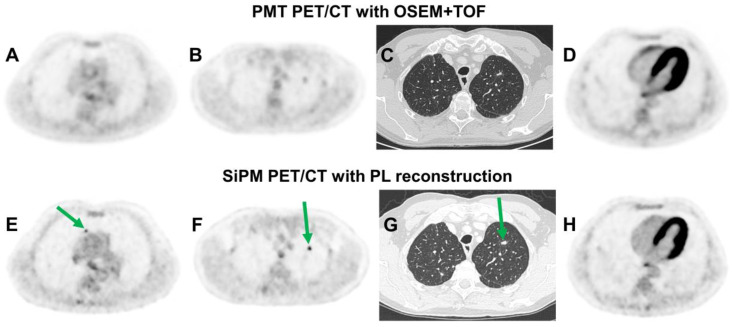
Images of two [^18^F]FDG-PET/CT examinations in a 63-year-old man with hepatic and pulmonary aspergillosis. The earlier examination was performed with a scanner equipped with conventional photomultiplier tubes (PMT) and reconstructed with OSEM and TOF (**A**–**D**). The second examination after 5 months used a SiPM-equipped PET scanner and PL reconstruction with a penalization factor beta of 450 (**E**–**H**). Two pulmonary lesions that showed only moderate [^18^F]FDG uptake during the earlier examination (**A**,**B**) appeared substantially more intense on the second scan (**E**,**F**). However, uptake in hepatic lesions declined (not shown), and both pulmonary lesions were unaltered in the CT scan (**C**,**G**), which suggested that the higher conspicuity of the pulmonary lesions was a result of improved reconstructed spatial resolution and lesion contrast recovery (CR) with the SiPM scanner and PL reconstruction. Based on phantom measurements, reconstructed spatial resolution was estimated at 7.8 mm full width at half maximum (FWHM) with the PMT scanner and 4.7 mm with the SiPM scanner. The improvement in image sharpness can also be seen in the myocardium (**D**,**H**).

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
