# Peer review of "Influences on PET Quantification and Interpretation"

_diagnostics, 2022, doi:10.3390/diagnostics12020451_

Round 1

Reviewer 1 Report

As a review, first, the layout of the manuscript lacks logic, which makes it confusing to read. Second, in addition to summarizing the work of others, the review also needs to have its own opinions, which is another deficiency of this manuscript. The current manuscript still needs a lot of serious revision and improvement. Therefore, I do not support its publication in the journal of Diagnostics.

Author Response

We have added a new figure as Figure 1 that illustrates the logic structure of the article from patient preparation to image acquisition and image interpretation. It also provides the corresponding subsections of the article.

We prefer a neutral depiction of the evidence. As this review is neither a commentary nor a controversial article, we have purposefully left our personal opinion aside.

Reviewer 2 Report

In this review, researcher provide a comprehensive summary of important factors influencing quantification and interpretation with a focus on recent developments in PET technology, and it could contribution to the field.

Author Response

Thank you!

Reviewer 3 Report

This work summarizes the factors influencing PET quantification and interpretation of resulting images, with useful inclusion of latest PET detector technologies (SiPM) and image reconstruction techniques (PSF, PL). Substantial references for all discussed topics are provided.

The article could benefit from a dedicated Discussion and/or Prospects sections, however, this information is provided in all subsections, which are otherwise well structured. This is a useful up-to-date review, which can be accepted in present form.

Author Response

Thank you!

Reviewer 4 Report

The authors are describing a very important aspect of PET imaging: the influence of PET parameters to the images’ interpretation. One review about the PET parameters would be essential for the new radioligands currently going to the market.

Despite the importance of such subject, the authors present only one part of the real parameters’ influence.

Major remarks:

  • The authors have only taken the most classical PET tracer: FDG. However, most of the problem of PET image interpretation depend mainly on the PET ligand and its metabolism and immunogenicity. More the ligand is complex (antibody/immunoPET, peptide, nanobody,…), more the impact of immune system and pathology environment can change the specific uptake of the radioligands. The authors have totally put aside this aspect. I would add more details about it.
  • One of the most problematic point for PET imaging is also the modelization. In order to assess the Binding potential, the VT,… to clear the image from non-specific targeting, more and more research studies are focusing on proposing some mathematical equation to modelize the tissue distribution of the tracer. Several studies involving modelization have shown excellent result. I would strongly advise the authors to specifically discuss about it.

Minor remarks:

  • The language has to be improved.
  • In general, some image and representation would help to understand the different notion. For the Partial volume effect for example, I would suggest the authors to describe the problem via a figure. For new PET users, the description is too abstract.

Author Response

We clearly agree with the reviewer that numerous factors may greatly influence the individual biodistribution of various radiopharmaceuticals. In subsections 2.1 and 2.2, the best-defined factors in FDG-PET are addressed, because they are of highest importance for daily routine clinical care and can be taken into consideration when performing clinical PET examinations.

However, biodistribution itself is not the main objective of the current article. The article focuses on factors that influence our ability to accurately depict and interpret PET data (instead of factors that influence the radiopharmaceutical’s ability to illustrate certain physiological processes). Considering the complexity of this issue, it would be better addressed in dedicated reviews, and the authors don’t feel competent to elucidate some of the factors that the reviewer mentioned (e.g., immunogenicity and microenvironment).

We agree that modeling of pharmacokinetic processes using PET data is a highly interesting and evolving field of research. However, the approach of such methods is different from the aspects that are addressed in the current review. This review summarizes factors that should be considered when quantifying and interpreting clinical PET images. It focusses on issues that are established in routine clinical care, because we regard those factors as most relevant. In our understanding, modeling has previously been proposed primarily for a priori estimates of the tracer’s biodistribution in the sense of prediction, e.g., for drug development, for pharmacokinetic modeling of receptors or theranostic approaches. For the reasons given above, all these applications are beyond the scope of the article.

To clarify this, the scope of the current review is now underlined in the extended introduction section. Furthermore, subsections 2.1 and 2.2 have been extended to provide references to reviews that have addressed the issues highlighted by the reviewer.

The manuscript was reviewed again by a native speaker for language improvements.

We agree with the reviewer that a figure helps to explain the concept of partial volume effect. Two new Figures 3 and 4 were added, and subsection 2.3 was modified to improve readability.

Reviewer 5 Report

Thorough overview article. Good structure and nice overviews. 

Author Response

Thank you!

Round 2

Reviewer 1 Report

 The manuscript can be accepted in present form after revision by the author.

Reviewer 4 Report

I am not totally satisfied with the reviewers' response. Especially about the pharmacodynamic modeling. Clearly the authors should investigate more about the impact of VT, linked to the receptor distribution and the BP on image quantification.